# Tumor-Infiltrating Dendritic Cells: Decisive Roles in Cancer Immunosurveillance, Immunoediting, and Tumor T Cell Tolerance

**DOI:** 10.3390/cells11203183

**Published:** 2022-10-11

**Authors:** Theodora Katopodi, Savvas Petanidis, Charalampos Charalampidis, Ioanna Chatziprodromidou, Panagiotis Eskitzis, Drosos Tsavlis, Paul Zarogoulidis, Christoforos Kosmidis, Dimitris Matthaios, Konstantinos Porpodis

**Affiliations:** 1Laboratory of Medical Biology and Genetics, Department of Medicine, Aristotle University of Thessaloniki, 54124 Thessaloniki, Greece; 2Department of Anatomy, Medical School, University of Cyprus, 1678 Nicosia, Cyprus; 3Department of Public Health, Medical School, University of Patra, 26500 Patras, Greece; 4Department of Obstetrics, University of Western Macedonia, 50100 Kozani, Greece; 5Laboratory of Experimental Physiology, Department of Medicine, Aristotle University of Thessaloniki, 54124 Thessaloniki, Greece; 6Third Department of Surgery, “AHEPA” University Hospital, Aristotle University of Thessaloniki, 55236 Thessaloniki, Greece; 7Oncology Department, General Hospital of Rhodos, 85133 Rhodos, Greece; 8Pulmonary Department-Oncology Unit, “G.Papanikolaou” General Hospital, Aristotle University of Thessaloniki, 57010 Thessaloniki, Greece

**Keywords:** dendritic cells, immunosuppression, immunoediting, metastasis

## Abstract

The tumor microenvironment plays a key role in progression of tumorigenesis, tumor progression, and metastasis. Accumulating data reveal that dendritic cells (DCs) appear to play a key role in the development and progression of metastatic neoplasia by driving immune system dysfunction and establishing immunosuppression, which is vital for tumor evasion of host immune response. Consequently, in this review, we will discuss the function of tumor-infiltrating DCs in immune cell signaling pathways that lead to treatment resistance, tumor recurrence, and immunosuppression. We will also review DC metabolism, differentiation, and plasticity, which are essential for metastasis and the development of lung tumors. Furthermore, we will take into account the interaction between myeloid cells and DCs in tumor-related immunosuppression. We will specifically look into the molecular immune-related mechanisms in the tumor microenvironment that result in reduced drug sensitivity and tumor relapse, as well as methods for combating drug resistance and focusing on immunosuppressive tumor networks. DCs play a crucial role in modulating the immune response. Especially, as cancer progresses, DCs may switch from playing an immunostimulatory to an inhibitory role. This article’s main emphasis is on tumor-infiltrating DCs. We address how they affect tumor growth and expansion, and we highlight innovative approaches for therapeutic modulation of these immunosuppressive DCs which is necessary for future personalized therapeutic approaches.

## 1. Introduction

Dendritic cells (DCs) are highly specialized antigen-presenting cells (APCs) that are able to stimulate effector T cell differentiation and trigger naive T cell activation [1,2]. In homeostatic signaling, they also play a role in the induction and maintenance of immunological tolerance [3,4]. Their morphological and functional variety suggests a high degree of plasticity and they play vital roles in the induction and regulation of innate and adaptive immune response (Figure 1) [5,6]. APCs are typically recognized by their constitutive expression of the major histocompatibility complex (MHC) II and costimulatory molecules, which allows them to capture, process, and deliver exogenous antigen to T lymphocytes. B cells, macrophages, and DCs are typically regarded as the three main populations of APCs. DCs are essential APCs to the immune system [7,8]. Since their discovery in 1973 by the pioneering work by Ralph M. Steinman, DCs play a role in bridging the gap between innate and adaptive immunity, including activation of antitumor T cells and the development of immunological memory [9]. DCs arise from myeloid progenitors known as common myeloid progenitors (CMP). CMP cells are then divided into two subtypes. Expression of the transcription factor Nur77 promotes monocyte differentiation of CMP. They can be further subdifferentiated into monocyte DCs (moDCs) [10]. In the absence of Nur77, CMPs differentiate into common progenitor dendritic cells (CDP), from which plasmacytoid DCs (pDCs) and conventional DCs (cDCs) are formed. Differentiated cDCs are initially immature and need maturation cues (such as inflammatory cytokines or DAMPs/PAMPs) in order to completely influence their function in the immune response [11]. Upon maturation and activation, DCs exhibit accelerated lymph node migration, decreased phagocytosis, increased MHC and costimulatory molecule expression, increased cytokine secretion, and MHC and CCR7 expression [12]. Mature DCs are able to stimulate immature T cells and trigger the adaptive immune response as a result of the phenotypic modifications that take place during activation. Thus, DCs possess a special capacity to deliver tumor antigens to the draining lymph nodes and trigger T cell activation, and modern immunotherapy using DCs is necessary for T-cell-dependent immunity and ICB response [13]. Additionally, tumor-resident DCs are beginning to play a part in controlling the T cell response inside tumors during treatment [14]. These roles put DCs at the center of the antitumor T cell response and imply that controlling the biological activity of these cells is a workable therapeutic strategy to subtly encourage a T cell response during therapy.

## 2. Dendritic Cells in Tumorigenesis

Dendritic cells and, especially, cDCs can potently stimulate T cells with exogenous and endogenous antigens and control T cell survival, proliferation, and effector activity [16]. This distinct role of cDCs is critical in the control of cancer, as cDCs pick up antigens from tumor cells and present them to T cells within the tumor microenvironment (TME) or after migrating to tumor-draining lymph nodes [17]. Additionally, TIDCs may circuitously decrease adaptive immune responses mainly by Treg induction [18]. In vitro, cytokines such as TGF-b, IL-10, and IL-2 cause DCs to promote Tregs development. These cytokines have recently been shown to work in conjunction with cosignaling/surface molecules, such as inducible T cell costimulator ligands, PD-L1, CD80, and CD86 which prompt DCs to trigger Treg expansion and immunosuppression [19]. Furthermore, pan-cancer analysis of single myeloid cells from human cancer types show that LAMP3^+^cDCs are associated with tumor progression, immune tolerance, and metastasis [20]. In general, immune stimulation has traditionally been associated with mature DCs, whereas immunosuppression and tolerogenicity have been associated with immature DCs. However, the functional flexibility of DCs is thought to be complex due to their characteristics, such as activation status, maturity, and polarization in TME (Figure 2) [21]. As a result, it is challenging to draw any broad conclusions about the functional roles of DCs in the TME.

## 3. cDC1

Both lymphoid and non-lymphoid organs, including blood, contain the human cDC1 subpopulation. Expression of CD141, chemokine receptor XCR1, C-type lectin CLEC9A, and the cell adhesion protein CADM1 distinguishes this subset [23]. Additionally, it is becoming more and more obvious that cDC1s play a crucial part in preserving CD8^+^ T cell activity within malignancies. The architecture of immune cells is essential for efficient communication in non-tumor models of immunity and secondary lymphoid organs [24]. Particularly, it has been demonstrated that the location of T cells close to cDCs is essential for the activation of an adaptive immune response [25]. This is supported by the fact that the generation of cytokines by tumor cDC1s is crucial for immunotherapy. Adoptive cell therapy requires cDCs to produce CXCL9/CXCL10 for adoptively transferred T cells to penetrate tumors. In addition, the expression of the chemokine receptor XCR1 is linked to the phenomenon of cross-presentation, which is performed by this subpopulation of DCs [26]. The basic leucine zipper transcriptional factor ATF-like 3 (BATF3) and IFN-regulatory factor-8 (IRF-8) are the primary transcription factors (TF) required for the development of cDC1 [27].

## 4. cDC2

Myeloid cDC2 are the predominant population of myeloid cDCs seen in human blood, tissues, and lymphoid organs. They express CD1c, CD2, FcR1, SIRPA, and the myeloid antigens CD11b, CD11c, CD13, and CD33. In addition to CD1c (for humans) and CD11b (for mice), SIRPa (CD172a), which is expressed by cDC2, defines this subset [28,29]. Due to its heterogeneity, cDC2 also expresses additional markers in accordance with their localization, such as CD1a in the dermis and CD103 in the gut. Human cDC2 can be induced to produce large amounts of IL-12 and to function as effective cross-presenting cells [30]. In the majority of the settings examined, their capacity to manufacture IL-12 is higher than that of cDC1. They release IL-23, IL-1, TNF-a, IL-8, and IL-10, but type III interferon is regularly produced at low levels. Human cDC2 are effective at activating CD8^+^ T cells, Th1, Th2, and Th17 cells in vitro [31]. The regulatory roles played by cDC2 in immune system physiology appear to be numerous but frequent. These cells are powerful inducers of regulatory T cells in the colon and preserve tolerance in the liver [32]. Additionally, CD4^+^ naive T cells are stimulated to display gut-homing markers and to produce Th2 cytokines when stimulated with vitamin D3. Only the cDC2 is capable of producing retinoic acid in response to Vitamin D3 stimulation [33].

## 5. Plasmacytoid Dendritic Cells

Plasmacytoid DCs have been shown to originate from both myeloid and lymphoid progenitors [34]. Production of type I interferon is carried out by pDCs that mainly do not express the myeloid antigens CD11c, CD33, CD11b, or CD13, in contrast to myeloid cDCs [35]. Nevertheless, several studies show CD11c expression on pDCs [36,37]. The GMDP markers CD123 (IL-3R) and CD45RA, which are down-regulated when DC progenitors develop into myeloid cDC, are still expressed by them [38]. They express CD4 at a higher level than myeloid cDCs, like all human DCs do. Additionally, the pDCs primary role in Type I IFN synthesis, is tightly regulated by a variety of surface receptors. IL-6, IL-12, CXCL8, CXCL10, CCL3, CCL4 and other proinflammatory cytokines and chemokines are all released by pDCs. Moreover, pDCs can deliver antigens to CD4^+^ T lymphocytes via production of MHC class II and costimulatory molecules. Furthermore, pDCs’ Type I IFN production is often associated with antiviral responses and has been implicated in autoimmune disorder etiologies, consistent with pDC triggering tolerogenic responses [39]. In addition, pDCs may be crucial for tumor development. The majority of IFN-α produced by the human body is released by toll-like receptor (TLR)-stimulated activated pDCs [40]. Numerous pDCs infiltrate tumors in many different types of cancer, however, they do not respond to TLR stimulation and do not produce much or any IFN-α. Additionally, Tregs are recruited by tumor-infiltrating pDCs into the tumor microenvironment, suppressing the immune response and fostering tumor growth [41].

## 6. Mo-DCs

Monocyte-derived DCs (mo-DCs) originate when monocytes are stimulated in the circulation, and then develop into DCs, a separate subset of DCs that is implicated in inflammation and infection (Figure 3) [42,43]. In autoimmune diseases, their activation and function may differ. While mo-DCs can be detected mainly in blood and tissues under physiological conditions, they are most frequently produced during inflammation and have a high degree of plasticity and variety in their roles [44]. These functions are greatly impacted by the microenvironment that is regulated and created by local DCs [45]. Human CD14^+^/CD16^−^monocytes perform context-dependent tasks, such as extravasating into tissues and differentiating into monocyte-derived macrophages, which produce IL-10 and encourage tolerance. They may be drawn to areas of inflammation and damage, where they secrete proinflammatory mediators such as TNF-a and IL-23 [46]. A specific subset of mo-DCs exhibits the 6-sulpho LacNAc (SLAN) carbohydrate modification of PSGL-1 and is thus termed SLAN-DCs [47]. These subsets produce substantial amounts of TNF-a and IL-12p70 in response to stimulation with TLR ligands. Blood slan/M-DC8^+^ cells have been functionally characterized as potent proinflammatory cells which trigger growth, cytotoxicity, and the generation ofIFN-γ by NK cells [48].

## 7. DC-Dependent Immune Surveillance

Cancer immune surveillance is regarded as a crucial host defense mechanism in order to prevent carcinogenesis and preserve cellular homeostasis [50]. DCs play a key role in this mechanism by monitoring the process of the immune system to detect and destroy malignant and neoplastic transformed cells (Figure 4). Recently, Hedge et al. showed that DC paucity leads to dysfunctional immune surveillance in pancreatic cancer. In detail, scarcity of DCs favors the expansion of tumor-promoting Th17 immunity. Restoring cDCs in pancreatic cancer can enhance CD8^+^ T cell and Th1 activity and restores anti-tumor T cell immunity [51]. Furthermore, tolerogenic DCs (tDCs) and exhausted CD4, CD8 T, and NK cells play a key role in immune-suppressed TME in esophageal squamous-cell carcinomas (ESCC). These DC-related immunosuppressive mechanisms may be responsible for the failure of immuno-surveillance and contribute to the immune suppressive state and disease progression [52]. Similarly, bone marrow-derived DCs can enhance tumor surveillance by effectively killing T cell lymphomas after activation with IFN-γ and TLR ligands in culture [53]. This DC activity can be attributed to specific receptors present on the surface of DCs. For example, CD91 on DCs controls the immunosurveillance mechanism for nascent, emerging tumors. Specifically, the CD91 receptor is required to activate immune responses to growing tumors. Effector immunological responses are suppressed and tumor incidence and progression are accelerated in the absence of CD91. Additionally, tumors that develop in the absence of CD91 express neo-epitopes that have higher immunogenicity indices. It has been demonstrated that human CD91 polymorphisms that are predicted to impair ligand binding affect the antitumor immune responses in cancer patients [54]. Likewise, in lung cancer, tissue resident cDC1 are blocked from capturing and engulfing cell-associated antigens as mice lung tumors develop. As a result of tumor-mediated downregulation of the phosphatidylserine receptor TIM4, which is extensively expressed in healthy lung resident cDC1 cells, loss of phagocytosis is associated with this mechanism. The activation of CD8^+^ T lymphocytes that are specific for the tumor is impaired and tumor growth is accelerated by TIM4 receptor inhibition and conditional cDC1 deletion. As a result, TIM4 on lung-resident cDC1 participates in immune surveillance and its expression is reduced in advanced malignancies [55].

## 8. Dendritic Regulation of Immunological Memory

Immunological memory is the immune system’s capacity to rapidly and accurately identify an antigen that the body has previously encountered and to launch an appropriate immunological reaction in response. Typically, these are secondary immune reactions to the same antigen. DCs play a significant role in this mechanism by their ability to activate the adaptive immune system, and also play an important role in shaping and directing the innate immune response. DCs can release large amounts of cytokines that can activate innate immunity cells and trigger T cell infiltration to further sharpen the overall immune response. The surface protein TIGIT suppresses memory T cell activation by promoting the generation of mature immunoregulatory DCs [57,58]. The methods and variability behind the development of memory CD4 T cells are yet unknown. It has been discovered that specific subsets of DCs can control the differentiation of a specific T-helper (Th)-cell subset by influencing cytokine signals around CD4 T cells. It is still unknown though, whether and how the regulatory DC subsets can control the differentiation of Tm-cells. Xu et al. showed that regulatory DCs program the generation of IL-4-producing memory CD4 T cells with suppressive activity. These regulatory DC-programmed Tm cells suppress CD4 T-cell activation and proliferation in vitro via IL-10 and thus reveal a new way of negative immune regulation by memory T cells [59].

## 9. DC-Associated Immunoediting

Cancer immunoediting is the mechanism by which the immune system can control and stimulate the growth of tumors. Tumor growth occurs in three stages known as elimination, equilibrium, and escape. These stages witness the editing of tumor immunogenicity and the acquisition of immunosuppressive mechanisms that promote disease progression. By modifying their antigenicity or creating an immunosuppressive milieu, tumors can subvert host immunity and promote the development of less immunogenic tumors through the process of cancer immunoediting. T-cell-mediated immune surveillance is also controlled by DC-related immunoediting. Tumor immune escape within the pancreas and peritoneum, in the absence of immunoediting, triggered tumor expansion, and PD-1 or CTLA-4 checkpoint inhibition was unable to restore antitumor immunity. Instead, decreased CD8^+^ T-cell priming by type I conventional DCs (cDC1) was linked to tumor escape. CD40 agonist therapy that enhanced cDC1 cross-presentation promoted T cell priming and prolonged T cell responses by epitope spreading, which helped to restore immunologic control. These findings highlight limitations to cDC1-mediated T cell priming imposed by specific TMEs that must be addressed for successful combination of immunotherapies and show that immunological escape of highly antigenic malignancies can occur without immunoediting in a tissue-restricted way [60].

## 10. Tumor T Cell Tolerance

T-cell activation and tolerance are strictly regulated to preserve immunological tolerance to self-antigens while ensuring efficient removal of foreign antigens. However, in the case of carcinogenesis, the TME aids T cell tolerance, which contributes to uncontrolled tumor growth (Figure 5) [61,62]. Recently, it was determined that monocytic DCs, a cDC subset, help maintain immune tolerance with low metabolic activity and display very potent APC activity [63]. Additionally, mast cells can also prompt DCs to mediate allograft tolerance by producing TNFα and GM-CSF which triggers the migration and function of peripheral tolerogenic DCs [64]. In many cases, the immune system fails to eradicate established tumors partly due to the induction of immune tolerance within the TME. For example, systemic dysfunction of pDCs play a critical role in the progression of ovarian cancer via induction of immune tolerance. Tumor-associated pDCs produce decreased levels of IFN-α, TNF-α, IL-6, macrophage inflammatory protein-1β, and RANTES suggesting the existence of a paracrine immunosuppressive loop [65]. Autologous tolerogenic DCs (ATDCs) can also trigger T cell tolerance. The suppression of T cell proliferation and the growth of Tregs by secreted substances are characteristics of ATDCs. High levels of lactate produced by ATDCs influence T cell responses toward tolerance. Indeed, T cells absorb lactate released by ATDCs, which reduces their glycolysis. By lowering T cell proliferative ability in vivo, ATDCs encourage increased levels of circulating lactate and postpone graft-versus-host disease [66]. Furthermore, non-small cell lung cancer (NSCLC) cells modulate the development of human CD1c^+^ subsets mediated by CD103 and CD205 which trigger the induction of tolerogenic CD1c^+^ DCs and the expansion of an immune suppressive microenvironment that causes tumor cells to escape immune surveillance [67]. Tumor secreted factors like chemokines can also activate T cell tolerance. For instance, cancer cell-produced CCR2 orchestrates suppression of the adaptive immune response by reducing infiltration and activation of CD103^+^ cross-presenting DCs and cytotoxic T lymphocytes (CTLs) [68].

## 11. Tumor-Induced Hypoxic Immunosuppression

In a plethora of tumors, specific DC subsets can create a microenvironment that not only supports tumor growth and metastasis but also reduces potential immune response by establishing hypoxic immunosuppression [70,71]. These subsets include tumor-associated DCs (TADCs), pDCs, tolorigenic DCs, and altered phenotype cDCs which are present inside the TME [72]. Recently, Devalaraja et al. showed that tumor cell-produced retinoic acid instructs intratumoral monocytes to develop into immunosuppressive macrophages rather than immunostimulatory DCs, which aids tumor cells to avoid immune responses [73]. Furthermore, tumor-infiltrating PD-1^+^ DCs mediate immune suppression in ovarian cancer by suppression of T cell activity and decreased infiltration of T cells in tumor tissues [74]. DCs can also stimulate regulatory B cells to produce IL-10 and mediate immune suppression of antigen-specific CD8 T cells in NOD mice [75]. In many cases, DCs can also trigger hypoxia signaling in the TME. For example, hypoxia-driven immunosuppression by cDC2 and Tregs triggers the release of Granzyme B, CCL20, and CXCL5 creating a hypoxic pro-tumorigenic network in HCC [76]. Defective DCs inside clusters of quiescent cancer cells can also prompt hypoxia-induced programs. Depending on where they were located inside the tumor, DCs and quiescent cancer cells (QCCs) act as immunotherapy-resistant reservoirs by directing a local immunosuppressive environment that is hypoxic and inhibits T cell activity [77].

## 12. Dendritic Cells in Clinical Trials

Clinical evaluation of DC-related cancer immunotherapy trials showed promising results. DC-based vaccines use both cytokine-activated DCs and DCs that can detect tumor antigens by using tumor lysates [78,79]. Despite the fact that some DC-related trials might not clearly involve them, DCs are crucial to tumor immunotherapy treatment approaches (Figure 6) [80,81]. Many tumor-specific or tumor-associated antigens, including CMV pp65, telomerase, Her2, Wilms’ tumor 1, and others, have been targeted by DC vaccines in numerous clinical trials. Patients with glioblastoma were vaccinated using CMV pp65 mRNA-loaded DCs in two stage I clinical pilot trials. The overall frequencies of IFN^+^, TNF^+^, CCL3^+^ polyfunctional, and CMV-specific CD8^+^ T cells increased in patients who received this immunization, as did long-term progression-free survival along with overall survival [82]. Patients with high-risk acute myeloid leukemia (AML) usually have enhanced telomerase activity in leukemic blasts. The researchers discovered that human telomerase reverse transcriptase (hTERT)-expressing autologous DCs were possible in a stage II clinical investigation. In adult AML patients, vaccination with hTERT-DCs appears to be safe and may be linked to better recurrence-free survival [83]. In a stage II clinical trial, DCs electroporated with Wilms’ tumor 1 (WT1) mRNA were reported to be an efficient method for preventing or delaying AML relapse following standard chemotherapy with the production of WT1-specific CD8^+^ T cell response [84]. DC1 vaccination was a secure and immunogenic method used in the clinical trial anti-HER2 to stimulate tumor-specific T cell responses in HER2 positive breast cancer patients [85]. Recently, it was demonstrated that an autologous tumor lysate DC vaccination had the ability to stimulate T cells. In a stage III clinical trial, it led to a tumor-specific immune response and improved patients’ overall survival with metastatic colorectal cancer [86]. Moreover, in an I/II clinical trial investigation, autologous mature DCs pulsed with dead LNCaP prostate cancer cells together with concurrent treatment did not prevent the activation of certain anti-tumor cytotoxic T lymphocytes [87]. In the therapy of patients with advanced cancer, some autologous DCs produced ex vivo and pulsed with tumor antigens showed very modest potential. Autologous DCs pulsed with allogeneic tumor cell lysates in a stage I clinical trial showed that DC immunotherapy with allogeneic tumor lysates can be safe and practical in tumor patients [88]. In addition, the use of DCs in combination with the TLR-3 agonist poly-ICLC against metastatic or locally advanced unresectable pancreatic cancer is optimistic in DC-related trials. The population of tumor-specific T cells is significantly increased, according to the results [89]. Additionally, melanoma patients who received autologous monocyte-derived mRNA electroporated DCs combined with ipilimumab experienced long-lasting tumor responses in a stage II clinical trial [90]. In a stage I clinical trial, CCL21-expressing adenoviral vector was used to transduce DCs. This procedure led to systemic immune responses directed towards the tumor’s antigen, improved tumor CD8^+^T cell infiltration, and elevated tumor PDL1 expression [91].

## 13. Conclusions

The ability to target tumor DCs in the anti-tumor immune response is becoming more well acknowledged. Although DC targeting single agents have had very little effectiveness, combining standard drug chemotherapy with new immunotherapies is a promising line of research. Given the intricate nature of the interactions between the tumor and immune system, more investigation is undoubtedly necessary to fully comprehend the function of the tumor immune microenvironment as a whole. DCs serve as important sentinel cells. DCs identify antigens, convert them into brief bioactive peptides, and assemble certain MHC-peptide complexes before presenting antigens to T cells in order to educate naive T cells for adaptive immune responses. Not only are DCs capable of activating T cells, but they can also regulate immunological activation, repression, and memory. Consequently, DCs, the T cell mentors, play a crucial role in immunological defense, surveillance, and homeostasis (Table 1). Furthermore, growing evidence suggests that DCs are an important component of tumor immunity. Preclinical research and clinical trials have demonstrated the potent efficacy of DC-based tumor immunotherapy. To combat tumor heterogeneity-related drug resistance, DCs can selectively detect, process, and present a variety of heterogeneous tumor antigens and activate T cells. More insights into the mechanisms of DC biology will emerge if new tools and modern methods are applied to DC research. Future research can readily encourage the creation of novel DC-based tumor immunotherapy approaches, and we believe that DC-based tumor immunotherapy has enormous potential for a future cancer cure.

## Figures and Tables

**Figure 1 cells-11-03183-f001:**
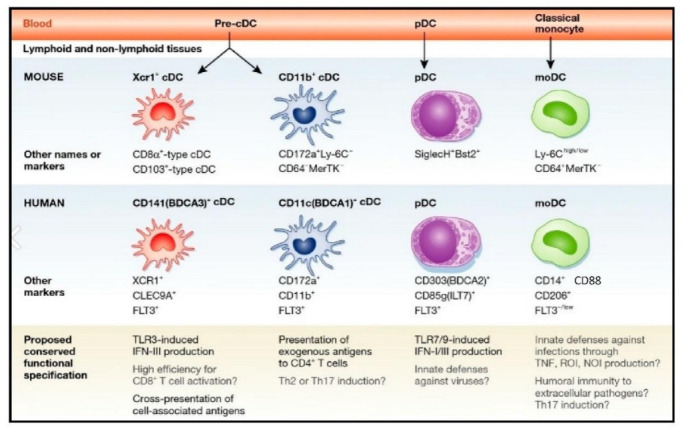
The major DC subsets in human and mouse. Each subset is characterized by specific surface markers and functional specification. Reproduced with permission from [15]. Copyright 2014, EMBO Press.

**Figure 2 cells-11-03183-f002:**
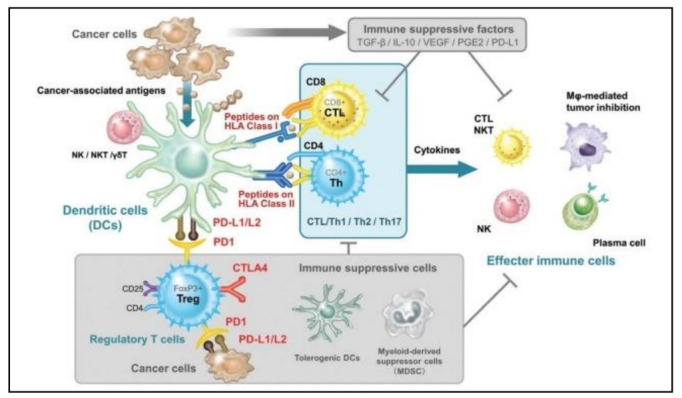
The dual immune stimulatory and immunosuppressive roles of DCs inside the TME. Reproduced with permission from [22]. Copyright 2016, Codon Publications.

**Figure 3 cells-11-03183-f003:**
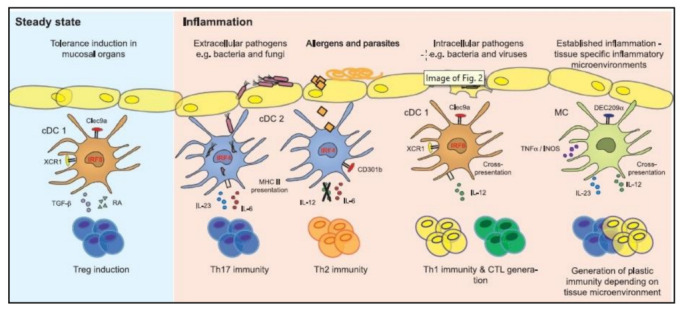
Diagram showing the functional characteristics of cDC subsets and monocyte-derived cell subsets in terms of their capacity to activate T cells during inflammation and the steady state. Reproduced with permission from [49]. Copyright 2015, Elsevier.

**Figure 4 cells-11-03183-f004:**
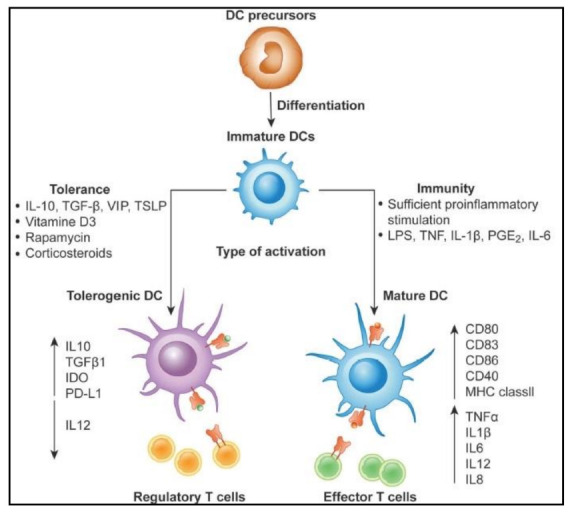
Differentiation of DCs is critical for efficient immune surveillance. Differentiation of tolerogenic vs. activated DCs generated from monocytes. In the presence of IL-4 and GM-CSF, DCs develop from DC precursors into immature DCs (iDCs). DCs become activated and change to a stimulatory phenotype in the presence of a maturation signal (proinflammatory cytokines and toll-like receptor ligands), which then triggers the development of effector/cytotoxic T cell responses. Reproduced with permission from [56]. Copyright 2019, Frontiers Media S.A.

**Figure 5 cells-11-03183-f005:**
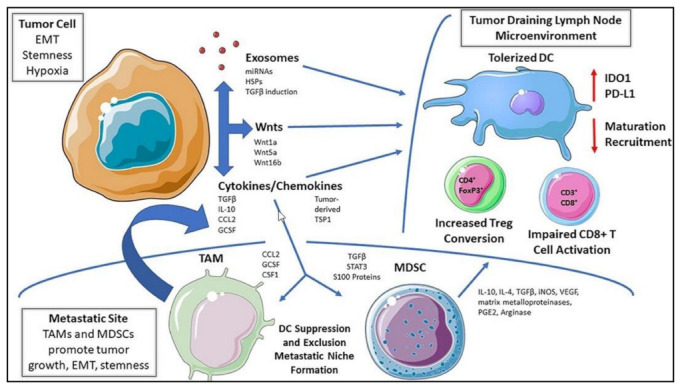
Mechanisms of DC tolerance in the tumor microenvironment. It is possible for tumor-derived soluble mediators to functionally tolerate DCs located inside tumor beds, tumor-draining lymph node tissues, or at more distant metastatic sites. This pathway promotes DC-dependent Treg differentiation while suppressing DC-mediated effector T cell responses, which aids in tumor development and progression. Reproduced with permission from [69]. Copyright 2019, Frontiers Media S.A.

**Figure 6 cells-11-03183-f006:**
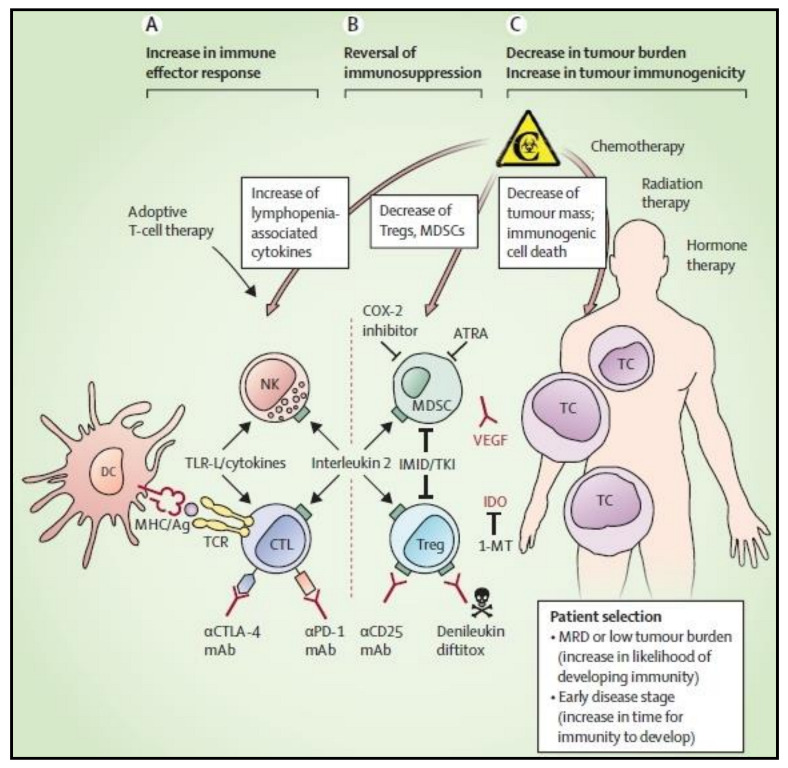
Multimodal techniques to enhance DC-based immunotherapy’s therapeutic effectiveness. DC-based cancer therapies aim to take use of DCs inherent ability to activate antitumor immune effector cells, such as natural killer cells and cytotoxic T lymphocytes that are specific to the tumor antigen. Dendritic cell therapy is being actively pursued in conjunction with therapeutic approaches that seek to strengthen the immune effector response (**A**), reverse cancer-associated immunosuppression (**B**), or reduce the tumor burden and raise the immunological susceptibility of the tumor cells (**C**). Reproduced with permission from [92]. Copyright 2014, Elsevier.

**Table 1 cells-11-03183-t001:** Representative table summarizing the role of DCs in each review section.

Mechanism	DC Role	DC Subsets	Cancer Type	References
**Tumorigenesis**	Treg expansion, immunosuppression, immune tolerance, tumor progression, metastasis	pDCs, LAMP3^+^cDCs,	Several cancer types	[19,20]
**Immune Surveillance**	expansion of tumor-promoting Th17, immune suppression in TME, enhancement/inhibition of immune surveillance	tDCs, BM-DCs, cDC1	Pancreatic cancer, ESCC, T cell lymphomas, Lung cancer	[51,52,53,55]
**Immunological memory**	Suppression of T cell activation, production of IL-4-producing CD4 Tm cells	Immunoregulatory DCs,	Several cancer types	[57,58,59]
**Immunoediting**	T-cell priming, epitope spreading	cDC1	Pancreatic adenocarcinoma	[60]
**T cell tolerance**	Immune T cell tolerance, tolerance-induced apoptosis, tolerance-induced T cell exhaustion	Merocytic DCs, pDCs, ATDCs, CD103^+^ DCs	C57BL/6 (B6) mice, Lung cancer, Breast cancer	[63,65,66,67,68]
**Hypoxic Immunosuppression**	Production of immunosuppressive macrophages, Granzyme B, CCL20 and CXCL5, hypoxia-induced programs	TADCs, pDCs, tolorigenic DCs	Sarcoma, HCC, TNBC	[72,74,75,76,77]
**Clinical Trials**	Stage I clinical pilot trials, stage II clinical trial, stage III clinical trial	mRNA-loaded DCs, (hTERT)-expressing autologous DCs, WT1 mRNA-DCs, autologous DCs	Glioblastoma, AML, Breast cancer, Pancreatic cancer, Melanoma	[82,83,84,85,86,87,88,89,90]

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
