# Peer review of "Tumor-Infiltrating Dendritic Cells: Decisive Roles in Cancer Immunosurveillance, Immunoediting, and Tumor T Cell Tolerance"

_cells, 2022, doi:10.3390/cells11203183_

Round 1
Reviewer 1 Report
This review, "Tumor-infiltrating dendritic cells. Decisive roles in cancer Immunosurveillance, Immunoediting and tumor T cell tolerance", by Katopodi et. al., covers a massive amount of information and data on dendritic cells (DC) in the tumor microenvironment (TME). The authors do an excellent job including both original publications and more recent publications, giving a broad and historical overview, and covering cutting edge research. Overall, sections such as the conclusions and introduction effectively wrap-up the subject matter. Language in many cases, is not succinct and could benefit from more substantial editing for clarity and less wordiness. The review is a good overview of the subject of TME dendritic cells, their role in immunosuppressive TMEs, and how they relate to tumorigenesis. The section on clinical trials is excellent and could benefit from a summary table. One area not thoroughly covered might be correlation Cancer Genome Atlas studies that find DC markers associated with tumor progression, for example, LAMP3+ cDCs described in Cheng et al. “A pan-cancer single-cell transcriptional atlas of tumor infiltrating myeloid cells.” Cell. 2021 Feb 4;184(3):792-809.e23. doi: 10.1016/j.cell.2021.01.010, or other TME DC research using this kind approach.
The weaknesses are the following:
1) The writing style often uses jargon and anthropomorphisms to describe mechanistic phenomena. It tends to be poetic at times, but not technical. For the review, those terms and phrases would be changed to be more precise and refer to mechanism. Some examples of this fuzzy language are below, under minor comments. I suggest changing those types of terms to be clearer to the reader who may not be familiar with the language, both those suggested examples, and throughout the manuscript in other instances not listed.
2) All figures are borrowed (with permission) from other published sources. This is a weakness as it tends to diffuse the review subject matter and make it more difficult to follow.
3) The authors’ review covers a massive amount of material that can be unwieldy for the reader. This reviewer strongly advises a Table (on tumor-infiltrating DC) summarizing each section of the review. This would aid the reader in finding the material and would add a specific figure directly from the authors.
An example layout is the following: a column header for TME DC function/area covered. For example, one row for Tumorigenesis (under that, cDc1, cDC2, pDC, etc); one row for Immune Surveillance (under that and indented, one example would be low DC numbers correlated with TH17 immunity [REF number, ie., 46] and another example, i.e., Decrease in CD91 DC Increased tumor Neo-antigens and Immunogenicity; one row for Regulation of Immunological Memory, and under that TIGIT expression, etc.; one row for Immunoediting; one row for T cell Tolerance; one row for Hypoxic Immunosuppression (under this, an example, Tumor-produced Retinoic Acid Increases Immunosuppressive Macrophage (67); and one row for Clinical Trials, where each trial approach was summarized, cancer type, reference, etc.
Note: There could be a column for: feature/observation, cancer type, reference number, etc.
Another suggestion would be to make the DC Clinical Trials section in a separate smaller table.
Minor Comments
Since there are no line numbers in the manuscript, I will list the section and phrase for the specific/minor comments.
pg. 1 “which is vital for tumor evasion of…”
pg. 2 “DCs are essential APCS to the immune system.” References could be added
pg. 3 “homeostatic” (not homeostasis) signaling
pg. 4 “cDCs are particularly skilled at stimulating”- “Skilled” is vague term – “potently stimulate” instead? Or something more specific
pg. 4 ”Which is one of the indirect methods" redundant, said 2x take out second time
pg. 4 “In vitro, cytokines like…” The word “like” is vague, use either “Such as”, or “including” or just strike the word “like.”
Pg. 5 “Due TO this”
Pg. 6 Very awkward sentence, here is a changed example, and some rewriting is needed for clarity: “Adoptive cell therapy requires cDCs produce CXCL9/CXCL10 for adoptively transferred T cells to penetrate tumors.”
Pg. 6 Line on cDC1 XCR1 expression is a little unclear.
Pg. 6 Made more succinct: “and IRF-8 are the primary…”
pg. 6 “Localization” better word than “placement”, or “tissue specificity”
pg. 6 Last sentence on page is vague and unclear to me. “These cells are (vs. have been said to) powerful inducers…”
pg. 7 “…and preserve tolerance…” (take out “to be in charge”)
pg. 7 redundant - change to “ Only the cDC2 subset is capable of…”
pg. 7 “…by plasmacytoid dendritic cells (pDC) that do not express…”
pg. 7 These lines are very wordy on pDC. I would suggest rewriting for succinct style. One sentence could be changed to, “The pDC primary role of Type I interferon synthesis, is tightly regulated by…”
pg. 7 Take out the words “ thanks to”, this is vague. Change to something like “ …to CD4+ T lymphocytes via production…
pg. 7 line on etiology should be made more succinct. It is unclear. One example is: “pDC Type I interferon production, often associated with antiviral responses, has been implicated in autoimmune disorder etiologies, consistent with pDC triggering tolerogenic responses (34).”
pg. 9 define ESCC
pg. 10 Typo “DCs surface”
pg. 10 The CD91 description could benefit from being explained more in the authors’ own words, to be clearer.
pg. 11 bottom/pg. 12 top. “…at the secondary, tertiary, and other levels” This levels statement is vague/not clear.
pg. 12 This statement is vague: “while instructing T cells to further sharpen the overall immune response.” Can you change this sentence to have a more clear mechanistic or functional explanation?
pg. 13 Define “epitope dissemination.” The sentence on CD40 agonist therapy can be made more succinct, and “regain immunologic control” be explained more specifically.
pg. 15 Define “License” in “ DCs can also license regulatory B cells to produce…” License is a specific immunologic term and needs definition for a general readership.
pg. 15 Is QCC= quiescent cancer cells? If so, define the acronym earlier.
pg. 16 “DC Cancer immunotherapy clinical evaluations have had positive outcomes.” More succinct wording may help clarify points.
pg. 16 “complete tumor lysates” is more of a method than what can be detected. Detection of all tumor antigens by using total tumor lysates?
pg. 16 Unclear statement “ While they may not explicitly use DCs in other DC-related trials…”
pg. 19 “Train” is a vague term. Perhaps “to train T cells, i.e., stimulate and activate T cells..”
pg. 19 Do you argue that the DC clinical trials have shown “great” efficacy? Some seemed modest efficacy.
pg. 19 Some words weaken the conclusions such as “into our underlying understanding”, “probably “ and “we think that”. Those words can come out.
Author Response
Response to reviewer comments:
We thank the reviewers for their interest in our manuscript, positive and helpful suggestions. Provided below is a point-by-point response describing our attempts to address, and when possible incorporate, all of their requested revisions in our manuscript.
Reviewer #1
Comment: The review is a good overview of the subject of TME dendritic cells, their role in immunosuppressive TMEs, and how they relate to tumorigenesis. The section on clinical trials is excellent and could benefit from a summary table. One area not thoroughly covered might be correlation Cancer Genome Atlas studies that find DC markers associated with tumor progression, for example, LAMP3+ cDCs described in Cheng et al. “A pan-cancer single-cell transcriptional atlas of tumor infiltrating myeloid cells.” Cell. 2021 Feb 4;184(3):792-809.e23. doi: 10.1016/j.cell.2021.01.010, or other TME DC research using this kind approach.
Response: We thank the Reviewer for his useful comment. The Cancer Genome Atlas studies that find DC markers associated with tumor progression, was added in the “Dendritic cells in Tumorigenesis” section of the manuscript and cited according to reviewer request.
Comment: The writing style often uses jargon and anthropomorphisms to describe mechanistic phenomena. It tends to be poetic at times, but not technical. For the review, those terms and phrases would be changed to be more precise and refer to mechanism. Some examples of this fuzzy language are below, under minor comments. I suggest changing those types of terms to be clearer to the reader who may not be familiar with the language, both those suggested examples, and throughout the manuscript in other instances not listed.
Response: We totally agree with the comment made by the Reviewer. Appropriate changes were made in the manuscript and terms and phrases were changed to be more precise according to reviewer`s request.
Comment: All figures are borrowed (with permission) from other published sources. This is a weakness as it tends to diffuse the review subject matter and make it more difficult to follow.
Response: We appreciate the comment made by the Reviewer. Since the figures are high quality and contain several complicated subfigures we tried to used several online creating figure software (e.g. Biorender). However permission for the usage of created figures requires annual subscription and limited usage (resctricted publication rights). Thus, all authors agreed on using preferred cited figures from selected publications. The preferred figures were used in order to illustrate and exemplify to the readers each subject topic.
Comment: The authors’ review covers a massive amount of material that can be unwieldy for the reader. This reviewer strongly advises a Table (on tumor-infiltrating DC) summarizing each section of the review. This would aid the reader in finding the material and would add a specific figure directly from the authors.
Response: We thank the reviewer for his useful comment. A Table on tumor-infiltrating DC, summarizing each section of the review was added in the manuscript according to Reviewer`s request.
Comment: An example layout is the following: a column header for TME DC function/area covered. For example, one row for Tumorigenesis (under that, cDc1, cDC2, pDC, etc); one row for Immune Surveillance (under that and indented, one example would be low DC numbers correlated with TH17 immunity [REF number, ie., 46] and another example, i.e., Decrease in CD91 DC Increased tumor Neo-antigens and Immunogenicity; one row for Regulation of Immunological Memory, and under that TIGIT expression, etc.; one row for Immunoediting; one row for T cell Tolerance; one row for Hypoxic Immunosuppression (under this, an example, Tumor-produced Retinoic Acid Increases Immunosuppressive Macrophage (67); and one row for Clinical Trials, where each trial approach was summarized, cancer type, reference, etc.
Note: There could be a column for: feature/observation, cancer type, reference number, etc. Another suggestion would be to make the DC Clinical Trials section in a separate smaller table.
Response: We appreciate the comment made by the Reviewer. For that reason, A Table with the above preferences was added in the manuscript according to Reviewer`s request.
Comment: Since there are no line numbers in the manuscript, I will list the section and phrase for the specific/minor comments.
- 1 “which is vital for tumor evasion of…”, Response: Appropriate changes were made.
- 2 “DCs are essential APCS to the immune system.” References could be added Response: References were added.
- 3 “homeostatic” (not homeostasis) signaling, Response: Appropriate changes were made.
- 4 “cDCs are particularly skilled at stimulating”- “Skilled” is vague term – “potently stimulate” instead? Or something more specific, Response: Appropriate changes were made.
- 4 ”Which is one of the indirect methods" redundant, said 2x take out second time , Response: Appropriate changes were made.
- 4 “In vitro, cytokines like…” The word “like” is vague, use either “Such as”, or “including” or just strike the word “like.” Response: Appropriate changes were made.
Pg. 5 “Due TO this”, Response: Appropriate changes were made.
Pg. 6 Very awkward sentence, here is a changed example, and some rewriting is needed for clarity: “Adoptive cell therapy requires cDCs produce CXCL9/CXCL10 for adoptively transferred T cells to penetrate tumors.” Response: Appropriate changes were made
Pg. 6 Line on cDC1 XCR1 expression is a little unclear. Response: Appropriate changes were made.
Pg. 6 Made more succinct: “and IRF-8 are the primary…”. Response: Appropriate changes were made.
- 6 “Localization” better word than “placement”, or “tissue specificity”. Response: Appropriate changes were made.
- 6 Last sentence on page is vague and unclear to me. “These cells are (vs. have been said to) powerful inducers…”. Response: Appropriate changes were made.
- 7 “…and preserve tolerance…” (take out “to be in charge”). Response: Appropriate changes were made.
- 7 redundant - change to “ Only the cDC2 subset is capable of…”. Response: Appropriate changes were made.
- 7 “…by plasmacytoid dendritic cells (pDC) that do not express…”. Response: Appropriate changes were made.
- 7 These lines are very wordy on pDC. I would suggest rewriting for succinct style. One sentence could be changed to, “The pDC primary role of Type I interferon synthesis, is tightly regulated by…”. Response: Appropriate changes were made.
- 7 Take out the words “ thanks to”, this is vague. Change to something like “ …to CD4+ T lymphocytes via production… Response: Appropriate changes were made.
- 7 line on etiology should be made more succinct. It is unclear. One example is: “pDC Type I interferon production, often associated with antiviral responses, has been implicated in autoimmune disorder etiologies, consistent with pDC triggering tolerogenic responses (34).” Response: Appropriate changes were made.
- 9 define ESCC. Response: Appropriate changes were made.
- 10 Typo “DCs surface”. Response: Appropriate changes were made.
- 10 The CD91 description could benefit from being explained more in the authors’ own words, to be clearer. Response: Appropriate changes were made.
- 11 bottom/pg. 12 top. “…at the secondary, tertiary, and other levels” This levels statement is vague/not clear. Response: Appropriate changes were made.
- 12 This statement is vague: “while instructing T cells to further sharpen the overall immune response.” Can you change this sentence to have a more clear mechanistic or functional explanation? Response: Appropriate changes were made
- 13 Define “epitope dissemination.” The sentence on CD40 agonist therapy can be made more succinct, and “regain immunologic control” be explained more specifically. Response: Appropriate changes were made
- 15 Define “License” in “ DCs can also license regulatory B cells to produce…” License is a specific immunologic term and needs definition for a general readership. Response: Appropriate changes were made.
- 15 Is QCC= quiescent cancer cells? If so, define the acronym earlier. Response: Appropriate changes were made.
- 16 “DC Cancer immunotherapy clinical evaluations have had positive outcomes.” More succinct wording may help clarify points. Response: Appropriate changes were made.
- 16 “complete tumor lysates” is more of a method than what can be detected. Detection of all tumor antigens by using total tumor lysates? Response: Appropriate changes were made.
- 16 Unclear statement “ While they may not explicitly use DCs in other DC-related trials…” Response: Appropriate changes were made.
- 19 “Train” is a vague term. Perhaps “to train T cells, i.e., stimulate and activate T cells..” Response: Appropriate changes were made.
- 19 Do you argue that the DC clinical trials have shown “great” efficacy? Some seemed modest efficacy. Response: Appropriate changes were made.
- 19 Some words weaken the conclusions such as “into our underlying understanding”, “probably “ and “we think that”. Those words can come out. Response: Appropriate changes were made.
Response: We thank the Reviewer for his useful comment. Appropriate changes were made in the manuscript according to Reviewer`s request.

Reviewer 2 Report
In their manuscript, Katopodi and Petanidis et al. have summarized the crucial role of conventional dendritic cells in cancer immune surveillance and immunoediting. DCs are indeed the crucial player in the regulation of tumorigenesis and thus are a focus of cancer immunotherapy. Overall, the authors have highlighted a vital topic; however, few concerns need to be addressed.
1- The manuscript is wordy and has grammatical errors. The reviewer suggests language editing.
2- The authors are suggested to redraft this statement for clarity ‘Selective immune escape within the pancreas and peritoneum, in the absence of immunoediting, trigger tumor expansion took place, and PD-1 or CTLA-4 checkpoint inhibition was unable to restore antitumor immunity.’
3- Plasmacytoid DCs have been shown to originate from both myeloid and lymphoid progenitors; therefore, it’s debatable. The authors must clarify and provide such details when describing the origin of pDCs in the manuscript with appropriate citations.
4- The reviewer disagrees which the authors’ statement, “Production of type I interferon is carried out by plasmacytoid dendritic cells. However pDCs do not express the myeloid antigens CD11c, CD33, CD11b, or CD13, in contrast to myeloid cDC.” Several studies that show CD11c expression on pDCS, for instance, PMID: 34608861 (Figure 2-Fig supplement 1).
5- In Fig.1, the authors should include CD88 as a specific marker for moDCs (PMID: 25769922).
6- Inconsistent use of ‘DCs’ and ‘dendritic cells’ was found in addition to random usage of acronyms without full forms.
